

# Adiabatic state distribution using anti-ferromagnetic spin systems

Koen Groenland[*]

**1** University of Amsterdam, QuSoft and Centrum Wiskunde en Informatica (CWI),
Amsterdam, The Netherlands

[*] k.l.groenland@cwi.nl

## Abstract

Transporting quantum information is an important prerequisite for quantum computers. We study how this can be done in Heisenberg-coupled spin networks using adiabatic control over the coupling strengths. We find that qudits can be transferred and entangled pairs can be created between distant sites of bipartite graphs with a certain balance between the maximum spin of both parts, extending previous results that were limited to linear chains. The transfer fidelity in a small star-shaped network is numerically analysed, and possible experimental implementations are discussed.

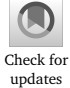

# 1 Introduction

Reliable transport of quantum states is essential for future quantum technologies [1]. For example, qubits stored within a quantum computer may need to be brought in close vicinity in order to perform quantum gates, and various (parts of) independent computers may need to be linked. If the quantum information is carried by a spin degree of freedom, then it is a natural choice to transport the states over a network of spinful particles [2,3].

In this work, we consider such a system of spinful particles, where the spins are coupled through anti-ferromagnetic (AFM) Heisenberg interactions. We show that, if Alice and Bob can adiabatically change the strengths of the couplings surrounding a small subsystem of a suitable network, then they can send each other quantum information and establish entanglement.

The protocols, as illustrated in Fig. 1, are then straightforward: to transfer a spin state, Alice starts uncoupled from the rest of the system and initializes her site in a state $\psi$. The rest of the system must be in a ground state with total spin $s = 0$. She then adiabatically ramps up some coupling to connect to the system, after which Bob ramps down the couplings connecting his site, finding $\psi$ at his now isolated site. Likewise, Alice and Bob can establish maximally entangled states between their sites by starting with the full network, including their sites, in a global $s = 0$ ground state. They then adiabatically uncouple their sites from the system, ending up with the unique $s = 0$ state shared between their sites. Such protocols have been abundant in existing literature (see Sec. 1.1), but were mainly focused on linear chains. Our results generalize these protocols to more general spin networks, and to more receivers in the case of transfer, under the assumptions given below.

The intuition that inspired this work is that the Heisenberg coupling preserves the total spin $\hat{S}^2$, and its $z$ component $\hat{S}^z$, of the whole system. If the final state has one part which is in a total spin 0 ground state, then the rest of the system must have copied whatever the initial spin properties were. The adiabatic theorem guarantees that, as long as couplings are changed sufficiently slowly with respect to a nonzero energy gap, the precise details of the procedure are unimportant. This reasoning is depicted in Fig. 1.

Our results require the following assumptions. We consider only bipartite graphs, and define $g$ as the difference between the maximum total spin of both parts. To transfer a spin-$s$ state, all parties must have $g = s$, and the system without either of the parties must consist of connected components, each of which must have $g = 0$. For entanglement distribution, both parties must hold a subsystem with a $g$ of opposite sign, and the global system must start connected and have $g = 0$. When the two parties are disconnected, the leftover system must again consist of components with $g = 0$ each. For either protocol, the precise details of the adiabatic path are unimportant, as long as the whole system satisfies a criterion we call *spin-s compatible* at all times, which guarantees the uniqueness of the ground state. We do not prove that these requirements are optimal, hence further generalizations may be possible.

## 1.1 Relation to previous work

Methods for state transfer through spin networks using unitary evolution can generally be categorized as either quenches, adiabatic evolution, or sequences of swaps. A quench involves a sudden change in a system's Hamiltonian, causing former eigenstates to evolve. Various constant interactions allow an excitation to to move between the ends of a linear chain with high fidelity, such as in the case of Heisenberg [4], $XY$ or fermionic hopping interaction [2,3]. Further engineering of these couplings can make the transfer perfect in theory [5–7]. Also, a sequence of SWAP operations between neighbouring qubits could be seen as a sequence of quenches.

In the following, we focus on adiabatic protocols, which exploit the adiabatic theorem to understand the evolution of the relevant eigenstate. Although these protocols are inherently

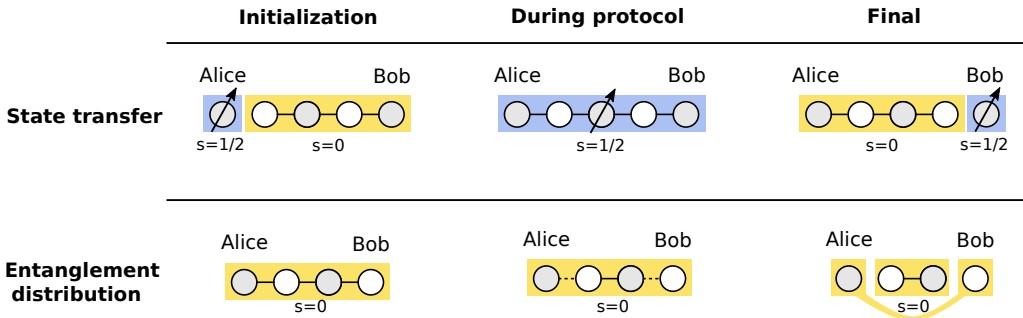

Figure 1: A sketch of the intuition behind our results. In the state transfer protocol, Alice initializes a spinful state while the rest of the system has zero total spin. The whole system then exhibits the same spin properties as the initialized state. After an appropriate adiabatic trajectory, Bob becomes disconnected. If the remainder of the system has a spin-zero ground state, then Bob's site must contain all spin information. Similarly, in entanglement distribution, Alice and Bob start with a state with zero total spin. After they disconnect, if the remainder of the system has total spin zero, then they must also share a total spin zero state.

slower than quenches, adiabatic protocols are often easier to implement experimentally because no precise timings are required, and because they are relatively resilient to decoherence and random or systematic errors in the control fields [8, 9].

Results can be further categorized by the type of quantum system (spins, electrons) and type of interactions under consideration. For our case, of spins with Heisenberg couplings, it was known that protocols of our type work on linear chains. Oh et al. [10] describe the state transfer protocol on three qubits with slightly more general XXZ interaction, and already note that the protocol seems to work on more general spin chain geometries. Agundez et al. [11] generalize the case of isotropic Heisenberg interactions to longer chain lengths, and describe superadiabatic optimizations. We note the closely related work by Eckert et al. [12], who consider a different SU(2)-symmetric interaction and give spin-conservation arguments, which inspired this work. The entanglement distribution protocol is closely related to the work by Campos Venuti et al. [7, 13], who describe the entanglement between the endpoints of the spin-$\frac{1}{2}$, finding stronger entanglement when the ends are more weakly coupled. Interestingly, the same system allows perfect quenched transport [7].

Slightly different spin models which allow similar adiabatic protocols are found in Refs. [9, 14, 15]. Moreover, much work has been done in the context of transferring particles rather than spins by adiabatically controlling hopping amplitudes [16–18], among which a broad class of protocols named *coherent tunneling by adiabatic passage* (CTAP) [19], which in turn can be applied to spin systems [20, 21].

Few of the transfer protocols deviate from treating a linear chain of sites, with notable exceptions being CTAP on engineered square and triangular grids [22] or with multiple receivers dangling along the linear chain [23], as well as spin transfer in a branched tree [24] or over multiple paths [25].

Our results specialize to adiabatic evolution in spin networks with Heisenberg coupling between neighbouring spins. We strengthen previous results by proving that protocols on a chain do indeed always work in the adiabatic limit, whilst extending the applicability to much more general network graphs. In particular, we allow different spins per site, we allow more general adiabatic paths, and sender/receiver are not limited to sit at the ends of the system. Moreover, we extend the state transfer protocol such that a state is not immediately transferred,

but rather encoded in the ground state of the whole system. After any amount of time, one out of various parties can then decide to localize the state at their site, using only local controls, without requiring any action from the other parties.

## 1.2 Document structure

The remainder of this paper is laid out as follows. Section 2 contains the main technical part of our work, as we make our intuition on the conservation of total spin more concrete, and we prove that given our graph restrictions, there is a unique ground state. Section 3 provides more details on our adiabatic protocols, section 4 discusses errors in real-world implementations, and section 5 addresses possible near-term experimental implementations. We finish with a discussion and outlook in section 6, and a conclusion in section 7.

# 2 Ground states of symmetry-protected subspaces

## 2.1 Preliminaries

Consider a network of spins, described by a graph $\mathcal{G} = (\mathcal{V}, \mathcal{E})$, with on each vertex (or site) $j \in \mathcal{V}$ a spin particle with total spin $s_j$, described by the spin operator $\hat{S}_j = (\hat{S}_j^x, \hat{S}_j^y, \hat{S}_j^z)^T$. Note that we allow a different value of spin $s_j$ per site. Spins which share an edge $(j, k) \in \mathcal{E}$ interact with isotropic, anti-ferromagnetic Heisenberg interaction of strength $J_{jk}$, and we assume full control over each of these interaction strengths. Such a system is described by the Hamiltonian

$$H = \sum_{(j,k)\in\mathcal{E}} J_{jk}\, \hat{S}_j \cdot \hat{S}_k, \qquad (J_{jk} > 0). \tag{1}$$

Throughout this work, we denote spin operators in upper case with a hat, and scalars as lower-case symbols without hat. We define the total spin $\hat{S}_{\text{tot}} = \sum_{j\in\mathcal{V}} \hat{S}_j$, such that $\hat{S}_{\text{tot}}^2$ has eigenvalues $s(s + 1)$, and total $z$-component $\hat{S}_{\text{tot}}^z = \sum_{j\in\mathcal{V}} \hat{S}_j^z$ which has eigenvalues $m$, taking on values ranging from $-s$ up to $s$ in integer steps. Likewise, for a subsystem $\alpha$ we denote the total spin operator on sites within that subsystem as $\hat{S}_\alpha = \sum_{j\in\alpha} \hat{S}_j$ with corresponding spin values $s_\alpha$ and $z$-magnetization $m_\alpha$. We use the term singlet to denote a state with $s = 0$.

Because $H$, $\hat{S}_{\text{tot}}$ and $\hat{S}_{\text{tot}}^z$ mutually commute, $s$ and $m$ can be used to index eigenstates of $H$. We let $V_{s,m}$ denote the subspace with fixed values $s$ and $m$. For systems with at least three sites, $V_{s,m}$ may consist of more than one state, hence we require a third quantum number to establish a complete basis. We denote the eigenbasis of $H$ as $|s, r, m\rangle$, where the label $r \in \{0, 1, 2, \dots\}$ orders the states within $V_{s,m}$ by *increasing* energy[1]. Fig. 2 graphically depicts this decomposition of the total Hilbert space.

We denote the $2s + 1$-dimensional spin-$s$ representation of SU(2) as $(s)$. It is known from representation theory that a system consisting of two spins $s_1$ and $s_2$ has its Hilbert space decomposed into irreducible representations of the total spin operator $\hat{S}_{\text{tot}}$ as

$$(s_1) \otimes (s_2) = \bigoplus_{s=|s_1-s_2|}^{s_1+s_2} (s). \tag{2}$$

The space of $n$ spin particles $s_1, s_2, \dots, s_j, \dots, s_n$ decomposes as

$$\bigotimes_{j=1}^{n} (s_j) = \bigoplus_s N_{s_1,s_2,\dots s_n}^s (s), \tag{3}$$

---

[1] The label $r$ is ill-defined whenever states have degenerate energies. This should cause no ambiguities in this work, as we consider only the ground state, which is assumed to be non-degenerate within the appropriate subspace.

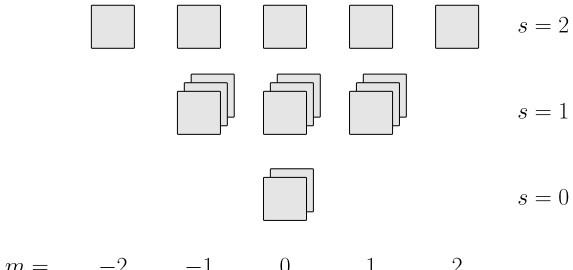

Figure 2: The decomposition of our Hilbert space using quantum numbers $s$, $m$ and $r$, here depicted for the case of four spin-$\frac{1}{2}$ particles. For each total spin $s = 0, 1$ or 2 (vertical), the $2s + 1$ possible $z$-magnetizations $m$ are laid out horizontally. The multiplicities of the spin spaces $N_{\frac{1}{2},\frac{1}{2},\frac{1}{2},\frac{1}{2}}^{\{0,1,2\}}$ are 2, 3 and 1, respectively, and states that differ by multiplicity-label $r$ are depicted in the depth-dimension. This way, each square corresponds to single state. The Hamiltonian $H$ must preserve labels $s$ and $m$, hence perturbing $J_{jk}$ can only excite states that differ in label $r$.

where the multiplicities $N_{s_1,s_2,\ldots s_n}^s$ of spin representation $(s)$ can be found by consecutive application of Eq. 2.

We say that the network graph $\mathcal{G}$ is bipartite if the vertices $\mathcal{V}$ can be split into two independent subsets $V_1$ and $V_2$ such that the couplings act only *between* the two subsets and never *within*, i.e. for all edges $(j, k) \in \mathcal{E}$, we must have $j \in V_1$ and $k \in V_2$. We say that two sites $j$ and $k$ are connected if there is a path of nonzero couplings between the sites, i.e. there exists a sequence $(J_{j,a_1}, J_{a_1,a_2}, \ldots J_{a_n,k})$ of nonzero elements. A graph or subsystem is connected if all pairs of vertices within that graph or subsystem are connected. If a system is not connected, then we use connected components to mean the largest possible subsystems in which all vertices are connected.

We define the spin imbalance $g$ of a spin system on a bipartite graph as the difference between the maximum allowed spin of each part:

$$g = \sum_{j \in V_1} s_j - \sum_{j \in V_2} s_j = \max s_{V_1} - \max s_{V_2}. \tag{4}$$

Note that spin imbalances can be simply added when combining spin systems: if subsystems have spin imbalances $g_1, g_2, \ldots$, then the combined system has a spin imbalance $g = \sum g_j$. A seminal result by Lieb and Mattis [26] states that each connected component with spin imbalance $g_j$ has a unique spin-$s_j$ subspace as ground subspace, whose the total spin $s_j = |g_j|$.

Our protocols critically rely on the adiabatic theorem [27], which states that a system remains in an instantaneous energy eigenstate, if the Hamiltonian is changed sufficiently slowly, and if there is a gap between the eigenstate's energy and the rest of the spectrum. In our case, we use the term gap to mean the energy difference between the ground state and the first excited state, within a symmetry-protected sector (typically $V_{s,m}$). Often, the gap may vanish in the thermodynamic limit, but we restrict ourselves to finite-sized systems. Still, for finite systems it is possible that the ground state becomes degenerate, in which case the gap closes and the adiabatic theorem can not be applied.

## 2.2 Preservation of the ground state of $V_{s,m}$

For our adiabatic protocols, we aim to show that information can be encoded in a protected subspace, in such a way that amplitude can not leak out of the subspace, and such that the subspace has a gap at all times. For the state transfer protocol, we aim to encode a quantum

state $|\psi\rangle \in \mathbb{C}^{2s+1}$ as $|\psi\rangle = \sum_{m=-s}^{+s} \psi_m |s, 0, m\rangle$, hence out of all the possible spin-$s$ subspaces ($s$) we require that one is the unique lowest energy subspace. Note that, for fixed $s$ and $r$ but varying $m$, all states have the exact same energy under $H$, guaranteeing that no relative phases occur within this subspace. Likewise, for entanglement distribution, we want to work within the unique global ground state which must have total spin $s = 0$.

In this section, we show the following:

1. The subspace $V_{s,m}$ is conserved under $H$ at all times.

2. On a connected and bipartite graph with $|g| \leq s$, the subspace $V_{s,m}$ has a unique ground state.

3. If a bipartite system is not connected, but consists of connected components with spin imbalances $g_1, g_2, \ldots g_l$, then the subspace $V_{s,m}$ has a unique ground state if $N_{|g_1|,|g_2|,\ldots,|g_l|}^s = 1$.

4. If one connected component has a spin imbalance $g = g_0 \neq 0$ while all other components have $g = 0$, then the full information encoded in the lowest-energy spin-$g_0$ subspace is accessible at the component with spin imbalance $g_0$.

We will argue why these observations hold in the remainder of this section, leaving in-depth discussion of our protocols for the next section. We stress that the requirements presented here are merely *sufficient* requirements and by no means the most general *necessary* conditions possible. By separating these conditions from the protocols, future generalizations can be straightforwardly applied to the protocols.

**1. The subspace $V_{s,m}$ is conserved under $H$ at all times.** This property follows because each individual term $(\hat{S}_j \cdot \hat{S}_k)$ of $H$ commutes with $\hat{S}_{\text{tot}}$ and $\hat{S}_{\text{tot}}^z$, hence $H$ can not change the quantum numbers $s$ and $m$, not even if the $J_{jk}$ are time-dependent.

**2. For $s \geq |g|$, $V_{s,m}$ has a unique ground state.** The proof of this claim follows from the results by Lieb and Mattis in Ref. [26]. We crucially need two of their findings:

Assume the system is connected and bipartite, with spin imbalance $g$. Then the following holds:

1. Within a subspace of fixed $\hat{S}_{\text{tot}}^z = m_0$, there is a unique ground state.
2. If $m_0 \geq |g|$, then this unique ground state has $s = m_0$.

From these observations, it follows that if $m_0 \geq |g|$, the subspace $V_{s=m_0, m=m_0}$ has a unique ground state. What about the other spaces? Recall that for fixed $s$, all values of $m$ have the same energy. Hence if $V_{m_0, m_0}$ has a unique ground state, then all $V_{m_0, m}$ have the same energies and in particular also a unique ground state. We conclude that any $V_{s,m}$ with $s \geq |g|$ has a unique ground state, while we can not make general statements about total spins smaller than $|g|$.

Let us take a step back here. With the previous two points, we have shown that in all connected and bipartite systems with $s \geq |g|$, one may adiabatically tune $J_{jk}$ without exciting the ground state $|s, 0, m\rangle$ of $V_{s,m}$: The quantum numbers $s$ and $m$ can never be changed by $H$, and the adiabatic theorem says that quantum number $r = 0$ is approximately conserved. However, when the system becomes disconnected, these results no longer hold. We therefore carefully analyze what happens upon disconnecting parts of the system, and we give sufficient conditions that guarantee a unique ground state.

**3. On disconnected subsystems.**    First, let us define precisely what we mean with connecting and disconnecting [2]. We consider a system consisting of two subsystems $L$ and $R$. Let the Hamiltonian on the combined system be of the form

$$H' = H_L + H_R + \sum_i \epsilon_i \hat{S}_{l_i} \cdot \hat{S}_{r_i}, \tag{5}$$

where $H_L$ and $H_R$ act only on subsystems $L$ and $R$ respectively, whilst $l_i$ are sites in $L$ and $r_i$ are sites in $R$. With connecting, we mean that the all parameters $\epsilon_i$ initially start at 0, and at least one of the $\epsilon_i$ is adiabatically increased to some positive nonzero value, such that the combined system becomes connected. Likewise, if at least one $\epsilon_i$ is nonzero, we may disconnect $L$ from $R$ by adiabatically lowering all $\epsilon_i$ down to zero.

For disconnected systems, the uniqueness of the ground state of $V_{s,m}$ does not necessarily hold any more. One might naively think that upon sending all $\epsilon_i \to 0$, the adiabatic process is saved because of two effects: If some $\epsilon_i \neq 0$ then there must be a unique ground state, and if all $\epsilon_i = 0$ the two subsystems are completely disconnected, hence we may as well look only at the ground states in each subsystem individually. However, if the gap closes *asymptotically* as $\epsilon_i \to 0$, then our adiabatic trajectory potentially traverses a region with *infinitesimally* small gap, hence the adiabatic time scale blows up.

To avoid such divergences, we require that both the connected (some $\epsilon_i > 0$) and the disconnected (all $\epsilon_i = 0$) configurations have a unique ground state. Then, because the eigenvalues of $H'$ are continuous functions of $\epsilon_i$, asymptotic vanishing of the gap is ruled out.

We propose the following sufficient requirement that guarantees a unique ground state of $V_{s,m}$: all connected components, labelled by $1, 2, \ldots l$, must have spin imbalances $g_1, g_2, \ldots, g_l$ such that $N^s_{|g_1|,|g_2|,\ldots|g_l|} = 1$. We will henceforth call this requirement *spin-s compatible*. We prove its validity as follows: each component $j$ has a unique spin-$s_j$ ground subspace with total spin $s_j = |g_j|$. The degeneracy of these subspaces allows the combined system to configure itself in various possible total spin configuration, according to Eq. 3. However, if $N^s_{|g_1|,|g_2|,\ldots|g_l|} = 1$, then there is just a single way in which the global ground subspace can configure itself that is compatible with total spin $s$, hence the ground state of $V_{s,m}$ must be unique. In particular, this means that we may adiabatically connect and disconnect without disturbing the ground state of $V_{s,m}$, as long as a system is spin-$s$ compatible both before and after the (dis)connection.

**4. If only a single component has nonzero spin imbalance, then all ground subspace information is accessible there.**    Let $\{g_0, 0, 0, \ldots\}$ be the spin imbalances of the connected components of some system, such that the spin imbalance of the combined system is $g_0$. In general, it holds that $N^{|g_0|}_{|g_0|,0,0,\ldots} = 1$, which means that $V_{|g_0|,m}$ has a unique ground state. We know precisely what this ground state looks like: all $g = 0$ components are in their unique singlet ground state, while the component with $g = g_0$ is in the state $||g_0|, 0, m\rangle$. These states, with $m$ ranging from $-|g_0| \leq m \leq |g_0|$, span the ground subspace of the global system, and are completely determined by the component with nonzero spin imbalance. Any operations performed on the subsystem with $g = g_0$ are in one-to-one correspondence with changes in the global ground subspace, and vice-versa.

In summary, throughout this section we showed that if the system remains spin-$s$ compatible, then the couplings $J_{jk}$ of $H$ can be changed adiabatically without affecting a quantum state's amplitude on the ground state of $V_{s,m}$. Moreover, by changing the couplings $J_{jk}$ of a system with

---

[2]Note the difference between '(dis)connecting' (the procedure described here) and '(dis)connected' (a property of a graph), although our definitions are such that no ambiguities should occur for any English conjugation of these words.

total spin $s$ in such a way that all components have $g = 0$ except for a single component which has $g = s$, then the amplitudes of $V_{s,m}$ for all $m$ are locally available at the latter component.

## 3 Applications

We discuss two applications in the context of quantum information, which are based on the results from the previous section: sharing a quantum state among multiple parties, and establishing entanglement between two parties. Fig. 3 shows these protocols on example graphs. The protocols assume a network graph that is bipartite and connected, but couplings $J_{jk}$ could be brought down to zero to break the connectedness. In order to check the correctness of the protocols, one should check two aspects. Firstly, a system initialized with total spin $s$ should always have a unique ground state in $V_{s,m}$, which we enforce by requiring that the system is spin-$s$ compatible at all times. Secondly, the initial and final states should be well understood such that they provide the utility that we claim. We illustrate situations in which our requirements are not fulfilled in Fig. 4.

### 3.1 Sharing a quantum state between multiple parties, such that any party can access the state

Consider a setup where $\ell$ cooperating parties $p_1, p_2, \ldots, p_\ell$ hold subsystems of a graph $\mathcal{G}$. In case one party $p_i$ experiences an emergency, it needs access to the state $|\psi\rangle = \sum_{m=-s}^{s} \psi_m |m\rangle \in \mathbb{C}^{2s+1}$, preferably without requiring any activity of the other parties. Because cloning $|\psi\rangle$ is generally impossible, the best option is to find a way to share the state between the parties. A protocol that offers a solution is as follows:

- Initialization: Without loss of generality, assume initially $p_1$ holds $|\psi\rangle$ locally. The system must be configured such that $p_1$ is disconnected from the rest of the system, and $p_1$ fully determines the ground state of the full system. $p_1$ can now initialize its subsystem in the state $|\psi'\rangle = \sum_{m=-s}^{s} \psi_m |s, 0, m\rangle$.

- Forming a resource state: $p_1$ connects to the system, and any other connections must be made such that all parties are on the same connected component. The state $|\psi'\rangle$ is now encoded in the spin-$s$ ground subspace of this component.

- Finalization: To access the information, any party $p_i$ can adiabatically disconnect, in such a way that it localizes the ground subspace information at its subsystem.

This protocol generalizes the transfer of a quantum state between two parties located at the ends of a chain, such as considered in Refs. [10, 11].

Let us discuss the requirements for graphs that allow such protocols. Firstly, to localize the ground state information at a single subsystem, the most general requirement we found is that any disconnected party $p_i$ should have $|g_i| = s$ while all other connected components have $g = 0$. Then from this, we derive that all parties $p_i$ must have the same spin imbalance $g_i$ (with the same sign), which follows because $\mathcal{G}/\{p_i\}$ should have $g = 0$ for any party $p_i$. The same holds for the resource state, where the component containing the parties must have spin imbalance equal to $g_i$ while the rest of the components have $g = 0$.

In between the initialization, the resource state configuration, and the finalization, the only constraint is that spin-$s$ compatibility is preserved, which is a less stringent requirement. For example, there could be more than a single connected component with $g \neq 0$ as long as $N_{|g_1|, |g_2|, \ldots}^{s} = 1$.

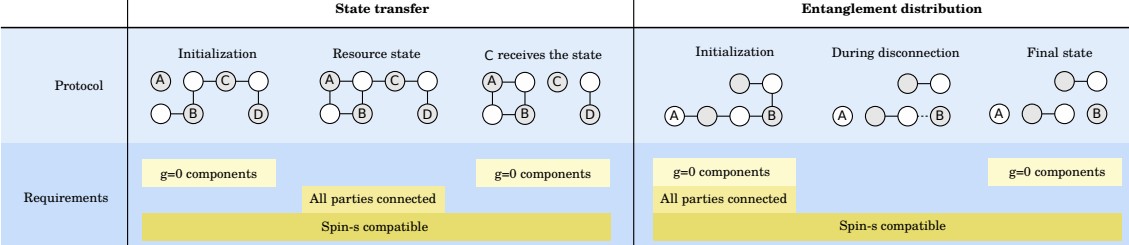

Figure 3: Example of a state transfer and entanglement distribution protocol, where we assume each site holds a particle with spin $s$. For readability, we label parties $p_i$ with capital letters. On the left, A initializes the system into a resource state encoding its original spin state, which is later obtained at C. On the right, the system is initialized in a global singlet, after which A and B disconnect to obtain singlet entanglement between their sites. At various stages in these protocols, the network must obey requirements such as (from top to bottom) all connected components except for the disconnected parties have $g = 0$, all parties must be connected, and throughout the whole protocol, the graph must remain spin-$s$ compatible.

An interesting situation occurs when one of the parties holds a subsystem which is not connected. In that case, transfer is still possible as long as $N^s_{|g_1|,|g_2|,\dots} = 1$ for a party whose connected components have spin imbalances $g_1, g_2, \dots$. If there are two connected components, this generally holds for any $|g_1 - g_2| \leq s \leq g_1 + g_2$. For three or more connected components, this condition can only be met when $\sum_i |g_i| = s$. Moreover, the combined spin imbalance must match the imbalance of the other parties, hence $|\sum_i g_i| = s$. We conclude that for more than two components, parties may have disconnected subsystems as long as all connected components have spin imbalances with the same sign (plus any number of $g = 0$ components), which properly add up to $s$.

## 3.2 Distributing maximally entangled singlet states

In this protocol, two parties $p_1$ and $p_2$ both hold subsystems on a graph $\mathcal{G}$. The ground subspaces of both parties are brought into the maximally entangled singlet state.

1. Initialization: The system must be in the unique ground state which has $s = 0$.

2. Finalization: Both parties are disconnected from the system, in such a way that the final state is a singlet on $\mathcal{G}/\{p_1, p_2\}$, in tensor product with a singlet on $\{p_1, p_2\}$.

Again, we turn to analyzing the requirements for allowed graphs. Firstly, to form a singlet state together, the parties $p_1$ and $p_2$ must have $|g_1| = |g_2|$. Because also $\mathcal{G}/\{p_1, p_2\}$ should have $g = 0$ and $\mathcal{G}$ must have $g = 0$, if follows that $g_1 = -g_2$.

In principle, the initial state allows any configuration with a unique spin-0 ground state. However, if $p_1$ and $p_2$ are restricted to controlling only the couplings directly surrounding their own subsystem, then the system must start such that the two parties are connected. For the final state, in order to be sure that the parties are not entangled with the rest of the system, the most general constraint we are aware of is that $\mathcal{G}/\{p_1, p_2\}$ must consist only of components with $g = 0$.

The precise trajectory of the coupling $J_{jk}$ is irrelevant as long as the system remains spin-0 compatible. Also, after the protocol is finished, individual connected components that remain can again be used as a starting stage for the same protocol. Notice that distributing entanglement between more than two parties in a single step is generally more complicated, because for multiple parties the spin-0 compatibility is easily broken.

Figure 4: Example issues that occur when graph requirements are not fulfilled. In the top row, party A cannot be disconnected in such a way that all other components have $g = 0$, eventually breaking spin-$s$ compatibility when the dotted couplings are set to 0. In the bottom row, the networks are such that A and B could in principle complete both protocols successfully, but during each protocol, disconnections are made such that spin-$s$ compatibility is broken. In each of these cases, the ground state of the relevant $V_{s,m}$ could become degenerate.

A closely related idea for entanglement generation was presented earlier in [13], where spins $p_1$ and $p_2$ sit at ends of a linear spin-$\frac{1}{2}$ chain, but $p_1, p_2$ are coupled more weakly to their neighbours than the spins in the bulk of the chain. By making this coupling ratio more extreme, the ground state was found to exhibit increasingly strong long-distance entanglement between the outermost spins. A later follow-up paper Ref. [7] investigated the usefulness of these outermost qubits in low-temperature chains for the teleportation protocol. Our results extend these earlier findings of long-distance entanglement to more general spin networks, and place them in a quantum control perspective.

# 4 Errors and scaling

So far, we dealt with the uniqueness of the ground state to prove that an adiabatic protocol is viable at *some* time scale, to which we remained agnostic. Moreover, we assumed perfectly sterile conditions: zero temperature and no interactions beyond those of Eq. 1. For real-world implementations, the actual scaling of protocol time and error susceptibility as a function of the number of sites $N$ would be of great importance, yet unfortunately, we are unable to give an in-depth and fully general characterization. Rather, this section collects known results related to this context, and numerically analyses the timing and errors in small systems.

Adiabatic processes are typically analyzed from the perspective of the energy gap $\Delta$, where the duration of the protocol $T$ is taken to scale as $T \propto \Delta^{-2}$ [8]. The Haldane conjecture states that linear Heisenberg chains consisting of particles with half-integer spin exhibit a $\Delta \propto 1/N$ gap, whereas integer spin particles feature a unique ground state with a constant gap [28]. However, care has to be taken that these results assume periodic boundary conditions which are not readily compatible with our protocols. For spin-1 particles, open boundary conditions give rise to four low-lying states separated from the rest of the spectrum by a constant gap. These lower states live in different spaces $V_{s,m}$, and their energies grow exponentially close in the thermodynamic limit [29].

The nature of the errors that arise during the protocol then depend on whether the system is truly $SU(2)$-symmetric: if it is, then only the gaps within the relevant $V_{s,m}$ are of any importance, and a system will not leave this subspace. However, to our best knowledge, all realistic systems described by Eq. 1 consist of spin particles whose magnetic moment interacts with magnetic fields $\vec{B}$, leading to interactions of the form $H = \sum_j \vec{B}_j \cdot \hat{S}_j$. Such fields break the $SU(2)$ symmetry, such that $V_{s,m}$ may no longer be conserved. However, if $\vec{B}_j$ is a constant function of $j$, then the protocols could still work, as we discuss in section 5.

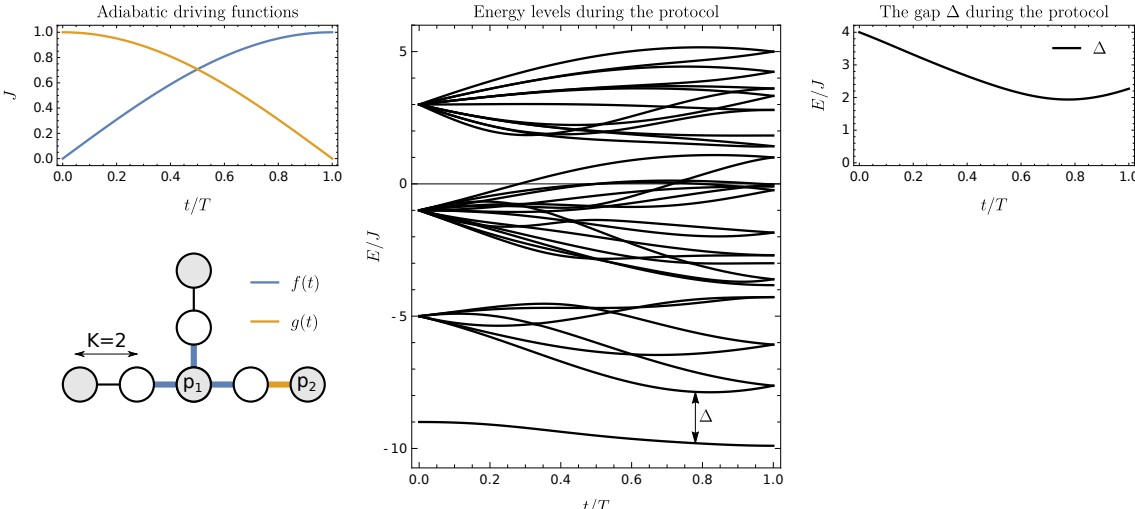

Figure 5: For the adiabatic transfer protocol in a system of size $M = 3$ and $K = 2$, we display the driving functions $f$ and $g$ and a sketch of the star-shaped system (left), the lowest 10 energy levels (middle) and the size of the energy gap (right) as a function of time. All units are normalized with respect to the protocol time $T$ and the coupling strength $J$.

Nonetheless, many results indicate that just the gap by itself does not necessarily reflect the viability of adiabatic protocols [30, 31]. Therefore, explicit numerical simulations of the full protocol appear to be the most informative. Such simulations have been performed for linear chains of spin-1/2 particles [10, 11], and for entanglement distribution on a spin-1/2 chain with the purpose of teleportation [7]. The latter work also calculates the gap in their system for lengths of up to one hundred. In Refs. [9, 12], closely related protocols are simulated.

Another important issue arises due to the fragility of Lieb and Mattis' statement that the ground state has definite total spin $s = |\max s_{V_1} - \max s_{V_2}|$, which is a fundamental building block of our protocols. Inhomogeneous magnetic fields of the form $B(\hat{S}^z_{V_1} - \hat{S}^z_{V_2})$ with amplitude $B = O(1/N)$ can cause the ground-state to have significant amplitude spread out over various spin sectors [32, 33]. Moreover, frustrated interactions such as $\hat{S} \cdot \hat{S}$ coupling between next-nearest neighbours may break the result of Lieb and Mattis.

We conclude that many threats can be identified, yet a general understanding of how these affect an adiabatic protocol is lacking. To give at least some insight in the practical performance of our protocols, we resort to numerical simulation. We select a concrete system that showcases our main contributions by allowing multi-party transfer and non-linear graph layout, namely qubits on a star-shaped graph.

### 4.1 Numerics on star graphs

We consider systems consisting of qubits ($s_j = \frac{1}{2}$) arranged in the shape of a star, where a center qubit is connected to $M$ arms each consisting of a linear chain of $K$ qubits, as depicted in Fig. 5. The total number of qubits is $KM + 1$. Note that such graphs reduce to a linear chain for $M = 1, 2$.

In the case of state transfer, the arm length $K$ must be even, allowing the center qubit to qualify as sender or receiver, as well as all qubits that are an even number of sites away from the center. We locate our sender $p_1$ at the center, and place receiver $p_2$ at the very end of the first arm. All couplings $J_{jk}$ are set to uniform strength $J$, except for the couplings connected

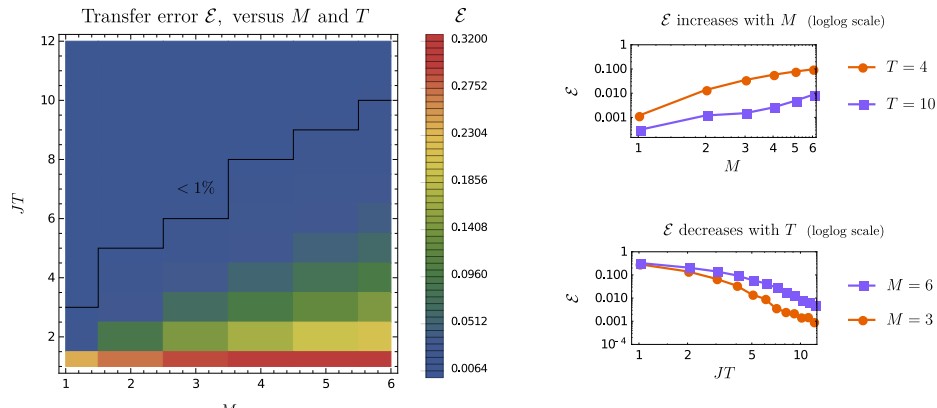

Figure 6: Transfer fidelity $\mathcal{E}$ for various choices of $T$ and $M$, for systems with arm length $K = 2$. For the small systems that we study, the fidelity seems to quickly converge to errors of less than 1%, indicated by the black zig-zag line, for protocol times just slightly larger than the intrinsic time scale $1/J$ of the system. On the right, cutouts of the main plot are displayed for fixed times (top) and fixed number of arms (bottom) on a log-log scale.

to $p_1$ or $p_2$, which we give time-dependent amplitudes $f(t)$ and $g(t)$ respectively. We choose

$$f(t) = J \sin\left(\frac{\pi t}{2T}\right), \quad g(t) = J \cos\left(\frac{\pi t}{2T}\right), \tag{6}$$

where $T$ is the total duration of the protocol.

Having defined our time-dependent Hamiltonian, we numerically solve Schrödinger's equation to find the unitary time-propagation $U_T$. As initial state we choose $|\psi(t = 0)\rangle = |\psi_0\rangle_{p_1} \otimes |0,0,0\rangle_{\mathcal{V}/p_1}$, where $|0,0,0\rangle$ is the global ground state with properties $s = 0, m = 0$. The state $|\psi_0\rangle_{p_1}$ is the state initialized by sender $p_1$, the choice of which does not influence the protocol's fidelity at this point, thanks to global $SU(2)$ symmetry. We then define the transfer error as

$$\mathcal{E} = 1 - \langle\psi_0| \quad \mathrm{tr}_{\mathcal{V}/p_2}\Big(|\psi(T)\rangle\langle\psi(T)|\Big) \quad |\psi_0\rangle,$$
$$\text{where } |\psi(T)\rangle = U_T|\psi(0)\rangle.$$

Here, $\mathrm{tr}_{\mathcal{V}/p_2}(\cdot)$ denotes the partial trace of the whole system except for site $p_2$. Fig. 5 depicts the driving functions $f$ and $g$, and the movement of the energy levels and the gap during the protocol.

Fig. 6 shows our results for the transfer fidelity for various protocol times $T$ and number of arms $M$, with arm length fixed to $K = 2$. For these small system sizes, very low error rates of less than 1% are readily obtained without further optimization of the protocol. Still, scaling up the system size by increasing $M$ clearly requires longer protocol times to achieve the same low errors. We leave the precise scaling of transfer fidelity for larger systems, possibly using optimizations beyond the adiabatic approximation, as an open question.

## 4.2 A numerical example where assumptions are violated

A minimal example of a transfer protocol that does not match our requirements is displayed in Fig. 7. The bottom-left panel shows a graph in which $p_1$ can disconnect while leaving all other components in $g = 0$, but $p_2$ cannot. Choosing spin-1/2 particles ($s_j = s = 1/2$) and the same time-dependent functions $f$ and $g$ for the couplings incident to $p_1$ and $p_2$ as before (Eq. 6),

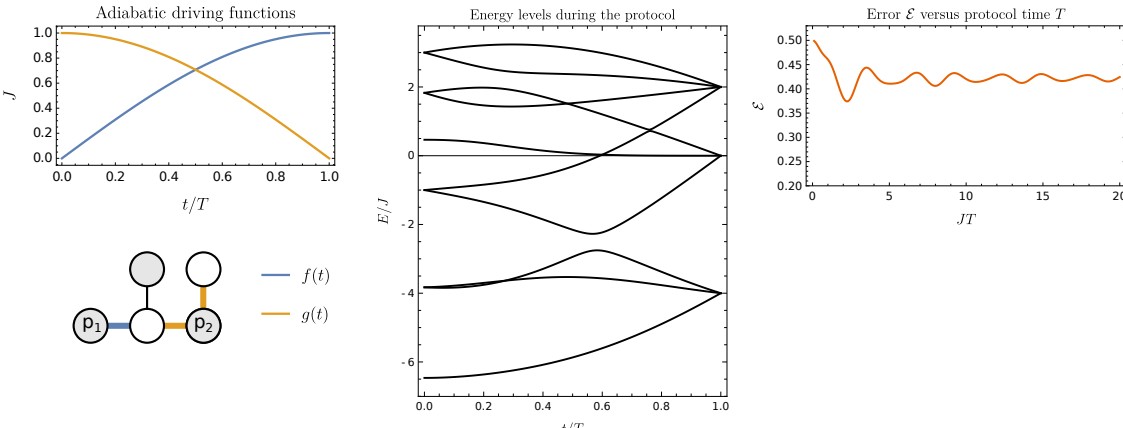

Figure 7: An example of a transfer protocol in which our requirements are violated. The left panel displays the graph layout and the time-dependence of the couplings connected to $p_1$ and $p_2$. Because $p_2$ is unable to disconnect while preserving spin-$s$ compatibility, there is a possibility that the relevant energy gap closes, which we indeed find here. As a result, the transfer error does not asymptotically decay to 0 as a function of $T$.

we calculate the energy levels over time, and the protocol error as a function of total protocol time $T$. Because the spin-$s$ compatibility is broken at the point $t = T$, there is no longer a guaranteed energy gap within $V_{s,m}$, and we indeed find that relevant gap closes precisely at time $T$. This has a devastating effect on the transfer error, which barely drops below 0.5, the latter corresponding to uniformly random outcomes at $p_2$. Importantly, the transfer error does not asymptotically decay to 0 as a function of $T$, a generic indicator that adiabatic transport fails.

# 5 Experimental implementations

The Heisenberg coupling of Eq. 1 can be approximated by systems forming a Fermi-Hubbard model in a regime of half-filling and strong on-site repulsion. Experimental platforms which have been proposed for quantum information processing in such regimes include ultracold atoms trapped in optical potentials [34, 35] and electrons trapped in quantum dots [36, 37]. Selectively varying the coupling between ultracold atoms requires delicate control over the trapping potential. In quantum dots, individual coupling strengths $J_{jk}$ are directly controlled using electronic gate voltages [38], making them a promising candidate for experimental implementation of our protocols.

Typical experiments will deal with a global magnetic field, adding a term of the form $\vec{B} \cdot \sum_{j \in \mathcal{V}} \hat{S}_j$ to the Hamiltonian. A field $\vec{B}$ oriented perfectly along the $z$-axis commutes with total spin operators $\hat{S}_{\text{tot}}$ and $\hat{S}_{\text{tot}}^z$, hence it does not change our conclusions on conservation of $V_{s,m}$ and the uniqueness of the ground state within these spaces. However, a relative dynamical phase between subspaces that differ in $m$ has to be accounted for. Moreover, it is no longer guaranteed that the global ground state is in spin sector $s = |g|$. More problematic could be magnetic noise which breaks the SU(2) symmetry. Minimizing the influence of such fields would be a major experimental challenge, and would form an interesting topic of further theoretical research.

# 6 Discussion and outlook

Given a physical system described by Eq. 1, there would be multiple ways to transfer quantum information, primarily through a quench [2], by a sequence of swapping operations, or by using the adiabatic steps that we propose. Our adiabatic approach has the disadvantage of having stringent cooling requirements and inherently slow dynamics. Moreover, it is unclear how the protocol time and error scale for large systems. On the other hand, adiabatic protocols have an advantage when it comes to control requirements, being relatively robust to decoherence and control errors [8,9], and requiring only a small number of couplings $J_{jk}$ to be adjustable as a function of time. The lower complexity of control makes such protocols worthwhile candidates for experiments on near-term quantum devices. We also note that our protocol could form a building block for an atomic swap operation that is repeated many times, allowing a trade-off between control complexity and time complexity, as the time of swap operations scales linearly with transfer distance.

Throughout this work, we remained agnostic with regards to the initialization of the ground state. Preparation can in principle be done by cooling, but having a system capable of adiabatically changing its couplings, it could be preferable to start from a simple initial state which has an adiabatic connection to the required ground state. Such a protocol was addressed in Ref. [9] for a spin-1 chain: start with a chain of sites with only the *odd* couplings active, such that the ground state is formed by two-site singlets. One can then adiabatically ramp up the even couplings to obtain the ground state of the fully coupled chain. Such initial states are either easier to cool, because their gap does not scale with the system size, or they may be prepared from a computational basis state using a quantum circuit of constant depth. Using our results, this initialization protocol readily extends to more general spin networks and any total spin $s$, with as only restriction that the system must remain spin-$s$ compatible during the process.

From a practical perspective, we note that many optimizations can be made to our protocol, most notably by reducing adiabatic errors such as discussed in Refs. [11,18]. On the theoretical side, we note that we have by no means exploited all the symmetries of Heisenberg systems yet. For example, we showed that *each* $V_{s,m}$ has a unique ground state, yet our protocols must stick to the global ground state due to our connection/disconnection procedure. Moreover, we proved that our requirements for a unique ground state are sufficient, but not that they are necessary; we expect that further generalizations are possible here, extending the applicability of adiabatic protocols. In a similar spirit, one might exploit ferromagnetic variations of the Lieb-Mattis theorem [39] if one circumvents problems arising from addition of spin quantum numbers. We believe that further examination of these open ends could lead to a better theoretical understanding of spin chains, and novel applications.

# 7 Conclusion

We extended previous work on two adiabatic quantum information protocols: one in which a spin-$s$ quantum state is adiabatically transferred to one out of many possible sites of a spin network, and another in which two parties extract entanglement in the form of a shared singlet state using a larger singlet state as resource. These protocols crucially rely on the preservation of subspaces $V_{s,m}$, and the uniqueness of their ground states. We hope that our methods could lead to other novel applications for manipulation of quantum information.

# Acknowledgements

Special thanks to Sjaak van Diepen for inspiring the ideas in this work. I would like to express my gratitude to Kareljan Schoutens and Jasper van Wezel for fruitful discussions and comments, to Freek Witteveen and Joris Kattemölle for aid with the mathematics of spin representations, and Bruno Nachtergaele for helpful comments on the manuscript.

**Funding information**  This research was supported by the QM&QI grant of the University of Amsterdam, supporting QuSoft.

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
