# Peer review of "Adiabatic state distribution using anti-ferromagnetic spin systems"

_SciPost Physics, doi:SciPost Phys. 6, 011 (2019)_

## Round 1 · Referee Report · Anonymous (Referee 1) · 2018-10-15

Strengths

1- Explanations are quite clear, including cartoon figures which really help with understanding 2- Nicely augments theory with numerical analysis 3- Thorough discussion of results, including what important issues could come up in the real world

Weaknesses

1- Although I am not going to require the authors to add this, it might have been nice to see some more cases analyzed numerically, including possibly cases which work but cannot rigorously be shown to work by the results in the earlier sections

2- There are some minor issues with some of the plots, as discussed in the requested changes

Report

The authors have done a detailed analysis of quantum state transfer and entangelment distribution on general graphs using sytems with anti-ferromagnetic Heisenberg interactions. The majority of the paper is devoted to discussing the theory around when this can and cannot be done, including both examples when it can, and counter-examples where some key assumptions do not hold and the transfer protocols do not work. The authors have also included a brief section with numerical analysis on a specific system, and discussion of possible experimental implementations. Overall, I think this paper is quite good, mathematically correct as far as I can tell, and represents a contribution to the field. I therefore think this paper should be published, with only minor changes, as discussed below.

The paper is clearly written and there are high quality cartoon figures to help explain the concepts.

minor changes:

The color bar on Fig. 6 is not labeled, while this is explained in the caption, it should also be labeled to make the figure more clear to the reader

Some energy levels are missing from the top of Fig. 5, while it is clear to a reader familiar with matrix numerics that more levels are not shown at the top, it would still make the figure more visually appealing to include all levels which would be visible on the plot.

I find it odd that the abstract only mentions some of the results, the discussion of experimental implementations and on different error types should be mentioned in the abstract, as these are important parts of the paper.

Requested changes

1- The color bar on Fig. 6 is not labeled, while this is explained in the caption, it should also be labeled to make the figure more clear to the reader

2- Some energy levels are missing from the top of Fig. 5, while it is clear to a reader familiar with matrix numerics that more levels are not shown at the top, it would still make the figure more visually appealing to include all levels which would be visible on the plot.

3- I find it odd that the abstract only mentions some of the results, the discussion of experimental implementations and on different error types should be mentioned in the abstract, as these are important parts of the paper.

---

## Round 1 · Referee Report · Anonymous (Referee 2) · 2018-11-29

Strengths

1 - The manuscript is written in a precise and clear language;
2 - It features an in-depth discussion over the necessary ingredients for carrying out a adiabatic quantum-state transfer and entanglement distribution in arbitrary graphs;
3 - It makes a significant contribution to the field of adiabatic quantum communication in general.

Weaknesses

1 - Although the work is pretty much self-contained, there is only one practical example by the end of manuscript;
2 - The literature of 'quantum communication in spin chains' (which is enormous) is not well addressed.

Report

The author outlines the key elements to perform adiabatic quantum communication tasks through a Heisenberg chain, wherein the involved parties are assumed to have
full control over a small set of couplings. Two protocols are particularly addressed, namely quantum-state transfer and entanglement distribution and a practical (numerical) example is provided using star graphs. The author also discusses limitations of the method and some possible experimental realizations.

I think the results are relevant and the theoretical reasoning behind them is very well presented. Also, it generalizes some previous results on adiabatic quantum communication protocols to arbitrary graphs. Therefore, for those reasons and given the overall quality of the work, I recommend it for publication.

I would only ask the author to: (1) explore a bit more the literature of
'quantum communication in spin chains' in the 'Introduction', that is to mention what are the main schemes put forward so far. A brief look at Refs [2] and [3] (and references within) is helpful; and (2) explain how exactly the results in Ref. [7] are generalized, as stated in Sec. 3.2.

Requested changes

1 - To explore a bit more the literature of 'quantum communication in spin chains' in the 'Introduction';
2 - To explain how exactly the results in Ref. [7] are generalized, as stated in Sec. 3.2.

---

## Round 2 · Referee Report · Anonymous · 2018-12-6

Strengths
1- Explanations are quite clear, including cartoon figures which really help with understanding
2- Nicely augments theory with numerical analysis
3- Thorough discussion of results, including what important issues could come up in the real world
Weaknesses
The main weaknesses of the paper have been addressed since the last report.
Report
I think the weaknesses I brought up before have been sufficiently addressed, it is understandable that the authors did not want to do too much more numerical work and I think there are still substantial results here. The additional numerics the authors did add have substantially improved the paper.
Requested changes
The authors have sufficiently addressed my comments and I now suggest publication without changes.

---

## Round 2 · Referee Report · Anonymous · 2018-12-19

Report
The manuscript has been improved and the author applied all the changes necessary. Therefore, I think it is now ready for publication.

---

## Round 2 · Author Response

Many thanks to the referees for their careful reports and useful comments. I am happy to implement the requested changes, which I think enhance the clarity and readability of the paper. The individual points are commented on below.
Report 1: 1. Fig. 6 now has more clear annotations.
-
The energy level plots (e.g. Fig. 5) now show all energy levels of the system. I did the same for the other example that was analyzed (see below).
-
Regarding the short abstract: The SciPost author guidelines state that "The abstract should fit within 8 lines of the template.". If the editor permits slightly surpassing this limit, I would like to update the abstract as submitted, which I hope strikes a good balance between conciseness and fully covering of the contents of the paper.
Lastly, regarding the recommendation to numerically analyze more cases, including those which do work yet do not satisfy the requirements stated in the paper: Unfortunately I was unable to find precisely such cases. However, I appreciate the idea of giving a better intuition by addressing different systems and depicting what happens in cases beyond the scope of this work. I chose to include a case which does not work, yet is "very close to working", namely a case where the 'spin-s compatibility' is violated when the receiver disconnects (section 4.2).
Report 2: 1. I expanded the discussion of previous results in the field of state transfer over spin chains. A new subsection 1.1 was added to collect these results.
- The last paragraph of Sec. 3.2 now discusses the connection between my entanglement distribution protocol, and the earlier result Ref. 7. Moreover, a reference to an earlier paper by the same authors, Campos Venuti et al. (2006) was also added.
I also appreciate the hint to include more concrete examples - I chose to numerically solve a case in which assumptions are violated, allowing a graphic presentation of how the transfer fails (section 4.2).

---

## Round 2 · List of Changes

- Added last line to abstract.
- Introduction: Re-arranged paragraphs such that first the protocol is explained, followed by the relation to existing literature. Added \subsections for the relation to previous work, and for the document structure. The literature discussion (now Sec. 1.1) was extended.
- 3rd paragraph of introduction: Added reference to Fig. 1 in the first line. Added lines at the end, which explain our contribution.
- Fig 4. caption is now more precise, mentioning explicitly that the ground state of V_{s,m} becomes degenerate.
- Fig 5. updated, now showing all energy levels.
- Fig 6. updated with better annotations.
- Sec. 3.2: Now explains the precise connection with the work of Campos Venuti at al. in the last paragraph.
- Sec 4.2: Added another numerical example, showing a case in which adiabatic transfer fails.
- Updated two citations that were recently published: Ban et al. (2018), Gratsea et al. (2018).

---

## Editorial Decision

published